# Improving the Ventilation of Machinery Spaces with Direct Adiabatic Cooling System

**Victor Mihai *** and **Liliana Rusu**

Faculty of Engineering, University "Dunarea de Jos" of Galati, 47 Domnească Street, 800008 Galati, Romania
* Correspondence: victor.mihai@ugal.ro; Tel.: +40-749065510

**Abstract:** Machinery spaces are provided with ventilation systems that ensure the necessary airflow for combustion and cooling. In some vessels, due to space constraints, the requested air flow for cooling cannot be achieved under extreme environmental conditions, and the engine load will need to be reduced. On the other hand, the outside air temperature can increase over 35 °C in some places, and the efficiency of the ventilation will be reduced. In these cases, other solutions for cooling the engine room should be analyzed. In this paper, the environmental conditions in the Romanian Danube area are analyzed to understand whether direct adiabatic cooling can be used to improve ventilation systems and what the challenges are after increasing the relative humidity inside the machinery spaces. Based on the data recorded, it was found that outside relative humidity substantially drops when the temperature increases, which ensures good conditions for the use of adiabatic cooling. This study demonstrates that by using direct adiabatic cooling, the air flow of the ventilation system can be reduced by more than 50%, which will reduce the pressure drop across the ventilation system, together with noise and energy consumption. After adiabatic cooling, the temperature and relative humidity inside the engine room will be fine for the functionality of the equipment, but the temperature–humidity index will be high, which means high discomfort for the crew. Therefore, it is concluded that this cooling solution should only be used in unattended machinery spaces.

**Keywords:** ventilation; relative humidity; adiabatic cooling; increasing efficiency; Danube River; ship; machinery spaces





## 1. Introduction

In general, the cooling of machinery spaces is performed by ventilation systems that use outside air, and in special cases, additional air coolers can be added. In order to have a ventilation system designed for all expected environmental conditions, the system is designed for extremely high temperatures for use on sailing vessels. The ventilation of the engine room is calculated according to the ISO 8861 standard [1], which states that 35 °C is required for the outside temperature and a temperature difference of a maximum of 12.5 °C, but the relative humidity influence is not considered [1].

Taking into consideration that the last time the size of the engine room was reduced and the total power was increased, some compromises were made during the detailed design and construction process of the vessel due to space constraints [2]. Consequently, the pressure drop across the ventilation systems can increase above the fan capacity, and finally, the air flow will be lower than the cooling requirements. Sometimes, the issue is discovered at a late stage of the project, during sea trials or in operation. In this stage of the project, in general, it is almost impossible to reduce the pressure drop. Therefore, other solutions are necessary to improve the ventilation system, such as Hendri et al. [3], who performed CFD analyses in relation to changing the air distribution and adding new fans. Unfortunately, this kind of improvement is expensive, time consuming, and sometimes requires big changes on board, which is not always possible. In other situations, the owner

might be looking for an easy and cheaper alternative; therefore, other solutions should be analyzed. One solution that can solve this issue is direct adiabatic cooling, which is easy to install and has low energy consumption, but it may have a negative impact on equipment and on people working inside if the relative humidity increases too much.

The efficiency of the system must be analyzed for each area based on the expected temperatures and relative humidity of the outside air.

According to the Mollier chart [4], the relative humidity of the air drops down when air temperature increases, and a high amount of energy can be exchanged using adiabatic cooling in hot dry air.

Considering that the necessary space for direct adiabatic cooling is low, this could be a solution to avoid reducing the capability of the vessel at extreme outside temperatures, but this solution could also be efficient for the normal operation of the vessel in hot and dry environmental conditions.

In order to check if direct adiabatic cooling can be used to increase the efficiency of the ventilation systems of the machinery spaces, the analysis proposed herewith will be done for a standard pusher used in the Danube area with propulsion power of about 3200 HP and two diesel generators, but only one running at 70 kW [5].

The main constraints and challenges are analyzed for direct adiabatic cooling, starting with the water quality of the local supplier and then energy analysis, up to the requirements for the system to avoid the spread of bacteria.

After analyzing the main constraints and challenges, the environmental conditions recorded during the summertime of 2021 are analyzed, and calculations are made to check the available capacity for adiabatic cooling for the engine room ventilation system. Based on all the data analyzed, it is concluded that the actual air flow of the ventilation system for the reference vessel can be reduced by 62% using direct adiabatic cooling in unattended machinery spaces.

The main objectives of the study are to check if direct adiabatic cooling is feasible for use in the engine room ventilation system, and to identify the challenges, constraints, and benefits of adding this cooling type to the engine room ventilation system.

## 2. Materials and Methods

### 2.1. Calculation of the Ventilation System

This was performed considering a pusher with propulsion power of $2 \times 1600$ HP used by the main River Sipping Company (Navrom) in the Danube area [5]. In this pusher, the electrical power can be provided by two diesel generators with a capacity of 77 kWe/generator, but only one diesel generator was running. The second one was a backup; therefore, it was not included in the calculation of the ventilation. The other equipment installed in the engine room has a low influence on the ventilation system.

The heat radiation for the main engines [6,7] and diesel generators [8,9], indicated in Table 1 above, is in accordance with the maker's data sheets. The heat radiation to the atmosphere indicated in the Caterpillar data sheets has a tolerance of 10% and is calculated for an air temperature of 25–29 °C. Heat radiation at 45 °C can be calculated using the dry exhaust and turbo manifold correction factor (DCF), which was developed by Caterpillar [10] using the fundamentals of heat transfer.

$$DCF = -0.011 \times \text{Ter} + 1.3178 \tag{1}$$

where "Ter" represents the Ambient Engine Room temperature [°C] and $DCF = 0.82$ [10].

**Table 1.** Heat radiating inside the engine room [6–9].

| Description | Quantity | Running | Heat Radiation | Total Heat Radiation |
|---|---|---|---|---|
| | | | | kW |
| Main engines | 2 pcs | 2 pcs | 101 kW/pc | 202 |
| Diesel generators | 2 pcs | 1 pc | 6.4 kW/pc | 6.4 |
| Exhaust pipe Main engine ND350 | 4 m | 4 m | 0.3 kW/m | 1.2 |
| Exhaust pipe Diesel generator ND125 | 6 m | 3 m | 0.15 kW/m | 0.45 |
| Electrical equipment | | | | 6.0 |
| Total Heat Radiated Inside the Engine Room | | | | Abt. 216 kW |

The heat radiated by the exhaust pipes is calculated according to the heat emission coefficients indicated in ISO standard 8861 [1]. The heat dissipation from the electrical equipment was estimated.

According to the engine maker data sheets, the following amount of air flow is needed for combustion, as shown in Table 2.

**Table 2.** Requested combustion air flow [6–9].

| Description | Quantity | Running | Combustion Air Flow | Total Heat Radiation |
|---|---|---|---|---|
| | | | $m^3$/h/engine | $m^3$/h |
| Main engines | 2 pcs | 2 pcs | 6650 | 13,300 |
| Diesel generators | 2 pcs | 1 pc | 350 | 350 |
| Total Combustion Air Flow $m^3$/h | | | | 13,650 |

According to the ISO8861 standard [1] the total air flow needed for the engine room ventilation should be at least the highest value calculated according to the following formula:

$$Qa = qh + qc \text{ (The air needed for cooling added to the air needed for combustion)} \quad (2)$$

$$Qb = 1.5 \times qc \text{ (150\% combustion air)} \quad (3)$$

where
$qh$–is the air flow needed for cooling,
$qc$–is the air flow needed for combustion.

The air flow needed for combustion for diesel engines is indicated in the engines maker data sheets, but if the maker data are missing, it can be calculated with the following formula:

$$qc = \frac{P \times mad}{\rho} \quad (4)$$

where
$mad$ = 0.002 kg/kW—is the air requirement for combustion [1];
$\rho = 1.13 \frac{kg}{m^3}$ —is the air density [1].
In our analysis, we used the maker data indicated in Table 2.

The air flow needed for cooling should be calculated using the following formula:

$$qh = \frac{\Sigma\Phi}{\rho \times c \times \Delta T} - 0.4 \times qc \tag{5}$$

where $\Sigma\Phi$—represents the sum of the heat radiation of equipment inside the engine room

$\rho = 1.13 \frac{kg}{m^3}$—represents air density;

$c = 1.01 \frac{kJ}{kgK}$—is the specific heat capacity of the air;

$\Delta T = 10\,°C$—represents the increase in the air temperature in the engine room.

The total air flow requested for cooling and combustion is presented in Table 3.

**Table 3.** Total air flow requested for cooling and combustion [1,6–9].

| Description | Qh—Cooling Air Flow | qc—Combustion Air Flow | Q—Requested Air Flow |
|---|---|---|---|
| | $m^3/h$ | $m^3/h$ | $m^3/h$ |
| $Qa = qh + qc$ | 54,520 | 13,650 | 62,710 |
| $Qb = 1.5 \times qc$ | | 13,650 | 20,475 |
| Requested Air Flow $m^3/h$ | | | 62,710 |

### 2.2. Existing Ventilation System on the Reference Vessel

The reference vessel is provided with a ventilation system, which contains 6 supply fans: 4 fans of 13,000 $m^3/h$ and 2 fans of 8000 $m^3/h$, total 68,000 $m^3/h$, which covers the air flow of 62,710 $m^3/h$ calculated for the ventilation system.

The ventilation system is provided with air supply fans only; therefore, the hot air from the room is naturally evacuated through the casing area in the upper part of the casing. According to the label on the fans, the electrical power of each fan is 2.2 kW.

### 2.3. Minimum Combustion Air Flow and Cooling Power Needed

As indicated in Table 3, according to the ISO 8861 standard [1], the minimum requested air flow is 20,475 $m^3/h$. Considering that the fans installed on the bord have no variable flow, one fan of 13,000 $m^3/h$ and one of 8000 $m^3/h$ must run to provide the minimum requested air flow.

The cooling capacity of the minimum air flow is calculated as follows:

$$\Phi m = (0.4 \times qc \times \rho \times c \times \Delta T) + (qha \times \rho \times c \times \Delta T) = \text{abt. } 40\text{ kW} \tag{6}$$

$$\Phi c = (0.4 \times qc \times \rho \times c \times \Delta T) = \text{abt. } 17.3\text{ kW} \tag{7}$$

$$\Phi cs = (qh \times \rho \times c \times \Delta Tc)\ \text{data processed from reference} \tag{7a}$$

Qha—air flow of ventilation system without combustion air;

$\Phi m$—cooling power for the minimum requested air flow;

$\Phi c$—cooling power for combustion air flow;

$\Phi cs$—sensible cooling power of cooling air;

$\Delta Tcs$ [k]—difference in temperature between outside and inside air;

$\Delta T = 10\,°C$ increase of air temperature in the engine room of 10 °C to keep the temperature inside the room below 45 °C, also when outside temperature is 35 °C.

Based on the total heat radiation inside the engine room of about 216 kW, calculated in Table 1, and the cooling power provided by the combustion air flow, the additional cooling power needed is about 199 kW.

### 2.4. The Main Aspects and Constrains of Ventilation Systems Related to Relative Humidity

The machinery and electrical equipment installed on board of the ship are designed for proper operation, even with an air temperature of 55 °C, according to IACS UR M40 [11].

Regarding the power of the main and auxiliary internal combustion engines, according to the International Association of Classification Societies (IACS), M28 "Ambient reference conditions" [12], they should be designed for an air temperature of 45 °C and a relative humidity of 60%. The inside temperature of 45 °C is actually the temperature used to calculate the ventilation system according to the ISO 8861 standard. These conditions represent the "absolute limits for humans to survive" [13] because the temperature humidity index (THI) for an air temperature of 45 °C and a relative humidity of 60% is over 100, according to Bohmanova et al. [14] and Termotecnica [15].

The temperature humidity index can be calculated according to the following formula [16,17]:

$$\text{THI} = (1.8T + 32) - (0.55 - 0.0055 \times U)[(1.8T + 32) - 58)] \tag{8}$$

where

    T [°C]–air temperature
    U [%]–relative humidity
    THI < 65—comfort state
    THI = 66–79—alert state
    THI = 80–100 discomfort state
    THI > 100 human body overheating leading to death

In the case of engine rooms with high temperature and humidity, Orosa et al. [18] highlighted that it is a good improvement for the comfort of the crew to have an intermediate room between the engine room and the control room or other room to reduce thermal shock after passing through the engine room. This intermediate room should be arranged at the concept stage of the project.

Referring to the influence of relative humidity on the efficiency of internal combustion engines, Rakopolous [19] concluded that high temperature and humidity will increase fuel consumption, but reducing temperature and increasing humidity will have a positive effect by substantially reducing pollution (NOx).

Based on the results of an analysis, Gayan [20] confirmed that reducing the temperature and humidity will increase the combustion efficiency and that the temperature reduction at 6 °C is "much more prominent than the increase in humidity by 28%" in the efficiency of the combustion process.

According to the International Council on Combustion Engines (CIMAC) [21], the increase in absolute air humidity will reduce the efficiency of the engine by "slowing down the combustion speed as well as reducing the maximum combustion temperature". However, the positive effect of reducing NOx is also highlighted.

On the other hand, high humidity can generate condensation in the charge air cooler due to the high pressure after the turbocharger and the drop in temperature inside the cooler.

Regarding the airborne spreading of viruses, the increase in relative humidity will reduce the degree of infectivity, as indicated in Figure 1, according to Bill et al. [22] and Ross [23].

*2.5. Adiabatic Cooling–General Overview*

The adiabatic cooling process is based on water vaporization into the hot air, which reduces the air temperature and increases the relative humidity. Therefore, it is more efficient in the case of hot and dry air. The main systems that use adiabatic cooling are as follows:

Direct adiabatic cooling—the water is sprayed directly into the air, which needs to be cooled down. In general, it is used to cool down air temperatures for open or industrial spaces with a high amount of fresh (outside) air flow.

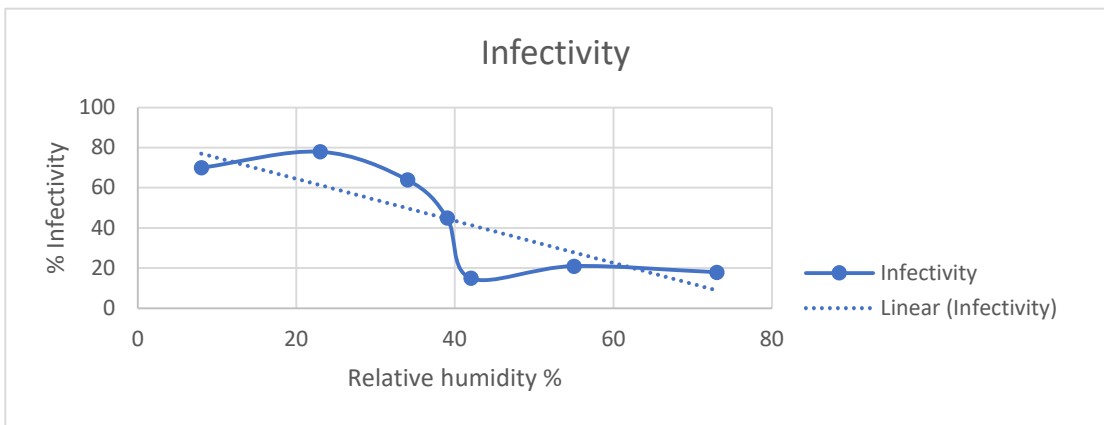

**Figure 1.** The degree of infectivity in relationship with the relative humidity inside the room (data processed from reference [23]).

Indirect adiabatic cooling—it is similar to direct adiabatic cooling in that the outside air used in the adiabatic cooling is separated by the inside air, which needs to be cooled by a heat exchanger, air to air type.

Adiabatic Cooling Towers—in this case, direct adiabatic cooling is used on the outside air, which is cooling down another fluid (water or other refrigerant) through a heat exchanger, air to water/refrigerant type.

The heat for vaporization of water liquid at 0 °C is about 595 kcal/kg [4] and 2500 kJ/kg according to the Engineering Toolbox [24]. Therefore, if the water is high pressure sprayed into the air with a relative humidity below 100%, it will evaporate by taking the energy for vaporization from the surrounding air. In this way, the air temperature will be reduced and the relative humidity will increase by adding the latent energy of water to the air.

The heat energy for vaporization of water valid for different water temperatures is indicated in Table 4 [4,24,25].

**Table 4.** Heat energy for vaporization of water [4,24,25].

| Temperature | [kJ/kg] | [Wh/kg] |
|:---:|:---:|:---:|
| 0 °C | 2500 | 695 |
| 20 °C | 2453 | 682 |
| 25 °C | 2442 | 678 |
| 30 °C | 2430 | 675 |
| 35 °C | 2418 | 672 |

In the analyses below the average water temperature of 30 °C and heat of vaporization of 675 W/kg will be used.

*2.6. The Main Constrains for Direct Adiabatic Cooling*

The fresh water that should be used in adiabatic cooling contains magnesium, calcium, and other particles that can be deposited and clogged in the water spraying nozzles.

In addition, when the water droplets evaporate, the minerals from the water that cannot evaporate will fall down or will be blown away by the air flow. Some of these minerals will go outside if there is a large air flow with high velocity inside the engine room, but others are expected to accumulate in different places inside the room.

To avoid this kind of issue, it is recommended to clean the water before using it for cooling. For this purpose, a reverse osmosis system that has good efficiency (over 95%) for removing salt and minerals from the water, with relatively low energy consumption, can

be used. Therefore, in the case of hard water with 1000 ppm, the water can be cleaned by a reverse osmosis system to 50 ppm.

Other solutions, such as chemical treatments for reducing the hardness of water, can also be adopted. In the case of the high-pressure cooling water mist system, to avoid clogging nozzles and high maintenance costs, Rasmossen [25] recommended keeping the water hardness used in this kind of cooling system at 70–90 mg/L (70–90 ppm).

Another issue that should be addressed in direct adiabatic cooling is bacteria, which can be contained in water. These bacteria are spread in the room through small droplets and aerosols. These aerosols, where bacteria are present, can be inhaled by the crew and go into the lungs, which can cause illness. Therefore, it is recommended that water be cleaned of bacteria and pathogens before using it for direct adiabatic cooling.

There are different pathogen's bacteria in water that can grow inside the system, like Legionella pneumophila, but in general the concentration is low, and the risk of disease will not occur if there are respected few general rules:

- To use clean water;
- To purge the system after using it to avoid water stagnation into the system;
- The water to be kept for less than one day in the water tank after cleaning.

Legionella pneumophila is multiplied at a water temperature of about 35–45 °C and it is killed when the water temperature is above 55 °C. UV light can be used to clean water from pathogens with low energy consumption.

Potable Water Quality and Hardness at Local Suppliers

According to the annual reports from the local water supplier [26,27] and other monthly reports [28,29], the requested and actual water quality of potable water in 2019 and 2020 was as follows (see data in Table 5):

**Table 5.** Fresh water quality on local suppliers [26–29].

| Parameter | Quality Requested by Actual Laws | Average Values in 2019 | Average Values in 2020 |
|---|---|---|---|
| Microbiological parameters–bacteria (pcs/100 mL) | 0 | 0 | 0 |
| Water hardness (Ca & Mg) [°G] | Minim 5 | 11–19 | 19–37 |
| Sulphates [mg/L] | 250 | 39–106 | 10–20 |
| Chlorides [mg/L] | 250 | 27–106 | 40–102 |
| Others have small quantity and are not considered | 0 | 0 | 0 |

Based on these reports it can be concluded that the average water hardness was medium in 2019 and mostly hard in 2020 [30] in some places. Considering that one German degree of water hardness is about 17.8 ppm (or mg/L) [31,32] the potable water from local suppliers can contain a quantity of Ca & Mg between 195 mg/L to 658 mg/L (ppm). Therefore, a cleaning or softening system should be used to reduce the water hardness to 90 ppm.

*2.7. Environmental Conditions in the Danube Area*

The analysis was done considering the worst-case scenario for the ventilation system, for an outside temperature above 25 °C.

The authors studied the time series of temperatures and humidity recorded in June, July, August, and September 2021 for the following places on the Danube area: Tulcea,

Galati, Calarasi, and Drobeta Turnu Severin. The data are available on the OGIMET [33] website, which was recommended by the Romanian Administration of Meteorology.

In Figure 2, the variation of temperatures is presented, and in Figure 3, the relative humidity for all the places analyzed corresponds to a period of 10 days at the end of July. As can be seen in this figure, the temperature variation is almost similar for all places with small differences, which can be between 0 and 5 °C. According to the data analyzed, the temperature for Calarasi can be considered as an average temperature for all the Romanian Danube area, with small differences in Galati and Tulcea, which are generally lower than 1 °C.

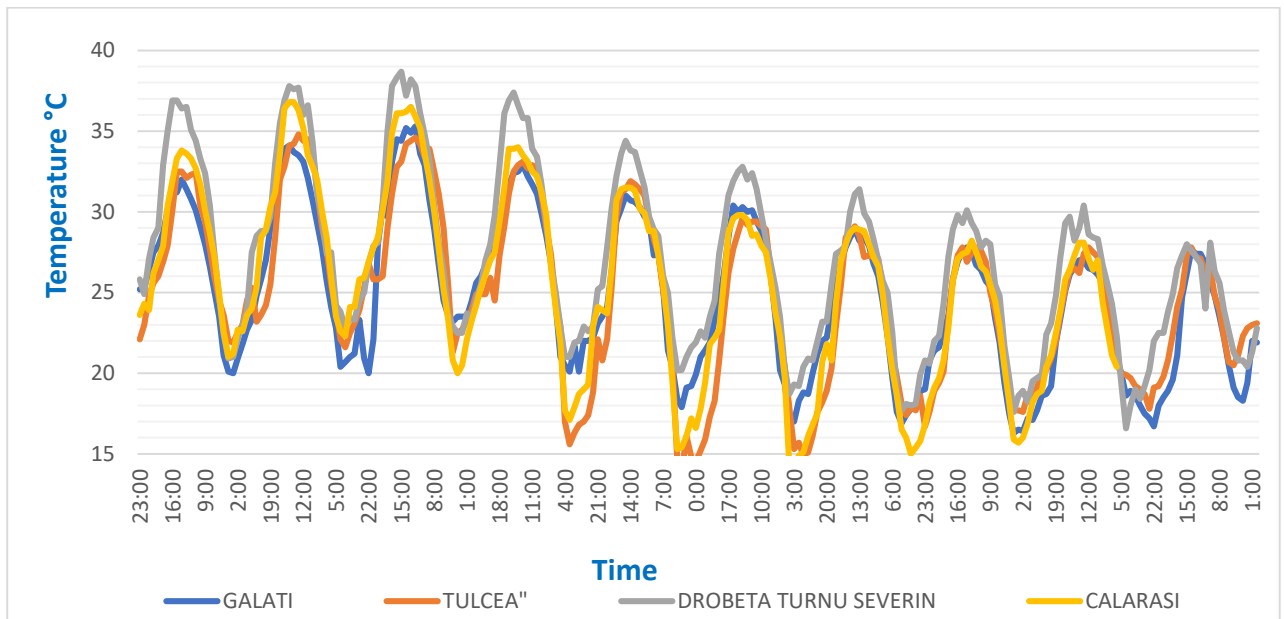

**Figure 2.** Time series of temperatures 21–30 July 2021 (data processed from reference [33]).

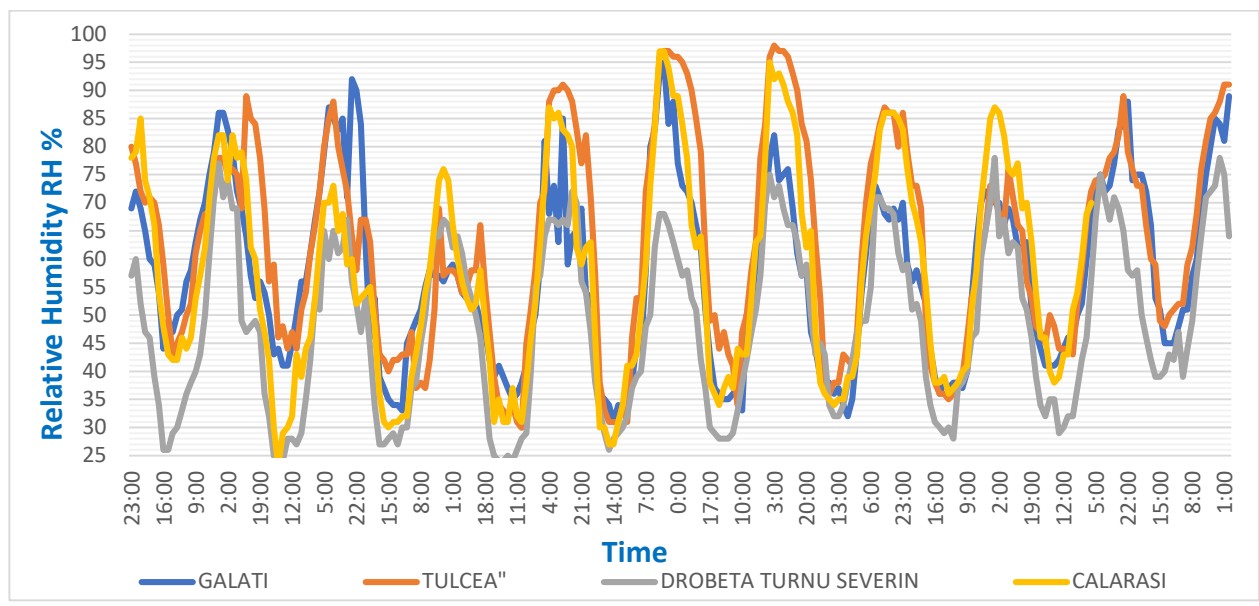

**Figure 3.** Time series of relative humidity 21-30th July 2021 (data processed from reference [33]).

In general, the difference in temperature between Calarasi and Drobeta Turnu Severin is higher and can be up to 3 °C, but in these cases, the relative humidity is up to 15% lower related to Calarasi. Therefore, based on the data analyzed, it was concluded that the

temperature and relative humidity valid for Calarasi can be considered average values for all places.

According to the data recorded in June, July, August, and September, it is concluded that the relative humidity is high during the night, when the temperature is below 25 °C, but is going down during the day when the air temperature is increasing, which creates good conditions for using the adiabatic cooling.

In general, the relative humidity was below 45% when the temperature was above 30 °C. The average relative humidity is 40–45% in July, and it is going down at the end of the month. In August, the average relative humidity was 35–45%, and it is going down at the end of the month. Maximum temperatures of 36 °C and 38 °C were recorded in July and August, respectively.

In September, the maximum temperature was 31 °C and the relative humidity had an average variation between 35% and 45%.

## 3. Results and Discussions

Based on the outside environmental data processed for June, July, August, and September, a temperature above 25 °C was recorded during 943 h. In the summary indicated in Table 6, it is concluded that the relative humidity is lower than 50% in more than 50% of the total time. It is higher than 60% in 16.5% of the total time, but it should be noted that in this time, the air temperature is in general below 30 °C. There is only 0.5% of the total time when the air humidity is higher than 60% and the outside temperature is over 30 °C.

**Table 6.** Time summary of relative humidity for temperatures above 25 °C in June, July, August, and September 2021 (data processed from reference [33]).

| Relative Humidity | All | RH < 40% | | 40% ≥ RH ≤ 50% | | 50% > RH < 60% | | RH ≥ 60% | |
|---|---|---|---|---|---|---|---|---|---|
| **Temperature** | ≥25 °C | <30 °C | ≥30 °C | <30 °C | ≥30 °C | <30 °C | ≥30 °C | <30 °C | ≥30 °C |
| June [hours] (Average temp. & RH) | 164 - | *1 (28 °C, 38%RH)* | *1 (33 °C, 39%RH)* | 27 *(30 °C, 45%RH)* | 29 | 36 *(30 °C, 55%RH)* | 22 | 45 | 3 |
| July [hours] (Average temp. & RH) | 337 - | 22 *(28 °C, 37%RH)* | 64 *(34 °C, 35%RH)* | 47 *(30 °C, 45%RH)* | 66 | 73 *(28 °C, 55%RH)* | 21 | 44 | 0 |
| August [hours] (Average temp. & RH) | 342 - | 62 *(28 °C, 37%RH)* | 63 *(33 °C, 35%RH)* | 54 *(30 °C, 45%RH)* | 39 | 49 *(27 °C, 54%RH)* | 11 | 62 | 2 |
| September [hours] (Average temp. & RH) | 100 - | 74 *(28 °C, 35%RH)* | 6 *(31 °C, 35%RH)* | 16 *(27 °C, 43%RH)* | 0 | 4 *(26 °C, 54%RH)* | 0 | 0 | 0 |
| TOTAL [hours] | 943 | 159 | 134 | 278 | | 216 | | 151 | 5 |
| | | 16.9% | 14.2% 293 31.1% | 29.5% 278 29.5% | | 22.9% 216 22.9% | | 16.0% | 0.5% 156 16.5% |

Note: The data indicated in the brackets "()" represents the average temperature and relative humidity according to the recorded data (e.g., in June the RH < 40%, and temp < 30 °C was recorded for 22 h, and the average temperature and relative humidity in this time was *28 °C, 37%RH*). The average temperature and relative humidity will be used for future calculations and analyses.

The cooling capacity of the air using direct adiabatic cooling by spraying the water at high pressure is analyzed below.

The cooling capacity can be calculated using the following formulas:

$$\Phi l = q_w \times \rho_w \times H_{va} \quad \text{cooling capacity of water spraying (latent heat exchange):} \quad (9)$$

$$q_w = \frac{\Phi l}{\rho_w \times H_{va}} \quad \text{waterflow for adiabatic cooling} \quad (9a)$$

where:

$\Phi l$ [kW] [kJ/s]—cooling power (latent heat of water spray vaporization)

$q_w \left[\frac{m^3}{s}\right]$—waterflow for adiabatic cooling

$\rho w \left[ \frac{kg}{m^3} \right]$—water density

$Hva \left[ \frac{kJ}{kg} \right]$—heat of vaporization of water (see Table 4, [4,24,25])

The total cooling capacity (latent & sensible heat exchange)

$$\Phi t = qha \times \rho a \times \Delta h \qquad (10)$$

With $\Delta h$ [kj/kg]—difference of air mixture (air &water vapor) enthalpy (Mollier chart [34])

$qha \left[ \frac{m^3}{s} \right]$—air flow of ventilation system without combustion air

$\rho a \left[ \frac{kg}{m^3} \right]$—air density

In all calculations, the air density will sometimes be considered according to the ISO standard [1], which will provide a spare in the cooling calculation related to the actual environmental conditions where the air temperature is lower than 35 °C.

Two alternatives for the actual ventilation system in the reference vessel are presented below in Table 7 and analyzed in Tables 8 and 9.

**Table 7.** The air flow, fans running, and load of the ventilation system in two alternatives analyzed for adiabatic cooling.

| Nr. Crt. | Fans Running | Total Air Flow m³/h | % Load of vent System | Combustion Air m³/h | Cooling Air m³/h |
|---|---|---|---|---|---|
| 1. | 1 × 8000 m³/h + 1 × 13,000 m³/h | 21.000 | 31% | 13,650 | 7.350 |
| 2. | 2 × 13,000 m³/h | 26.000 | 38% | 13,650 | 12.350 |

In the first alternative the cooling capacity of the ventilation system with direct adiabatic cooling is calculated for the minimum requested air flow, by starting one fan of 13.000 m³/h and one fan of 8000 m³/h (total 21.000 m³/h).

In the second alternative, the air flow is increased so that the requested cooling capacity can be covered in all outside environmental conditions without exceeding the relative humidity of 60%. In this calculation, two fans of 13.000 m³/h (total 26.000 m³/h), are used.

In Tables 8 and 9, columns 01, 02, and 03 are according to Table 6, based on the data recorded. The temperature and humidity index indicated in columns 04 and 13 are calculated according to formula (8) for the outside and inside air after water spraying.

The absolute moisture (column 05), enthalpy (column 06), and final relative humidity (column 12) are determined from the Mollier Chart [34]. The sensible cooling power of the cooling air (column 07) is calculated according to Formula (7a). The adiabatic cooling latent heat (column 08) is calculated as the difference between the total cooling power needed (199 kW) and the sensible cooling power calculated in column 07. The water flow for adiabatic cooling (column 09) is calculated with Formula (9a) using the latent cooling power from column 08. The absolute moisture added by the adiabatic cooling (column 10) is calculated according to the water flow in (column 09) and air flow for cooling. The absolute moisture after the direct adiabatic cooling is calculated as a sum of columns 05 and 10, and based on this, the final relative humidity is determined from the Mollier chart.

**Table 8.** THI index, relative humidity, and direct adiabatic cooling power calculated for cooling air flow of 7350 m³/h and total air flow of 21.000 m³/h (one fan of 8000 and one of 13.000 m³/h), inside air temperature 45 °C, RH according to calculation.

| | Outside Air | | | | | Cooling Power | | | | Inside Air | | |
|---|---|---|---|---|---|---|---|---|---|---|---|---|
| Hours | Temperature | RH | THI—Temp Humidity Index | Absolute Moisture | Enthalpy | Sensible Cooling Power | Adiabatic Cooling (Latent Heat) | Water Flow | Moisture Added | Absolute Moisture After Direct Adiabatic Cooling | RH | THI—Temp Humidity Index |
| **01** | **02** | **03** | **04** | **05** | **06** | **07** | **08** | **09** | **10** | **11** | **12** | **13** |
| | [°C] | (%) | | (g/kg) | (kJ/kg) | [kw] | [kw] | [L/h] | (g/kg) | (g/kg) | (%) | (%) |
| 1 | 28 | 38 | 74.1 | 8.9 | 49.9 | 40 | 159 | 236 | 28.4 | 37.4 | 60 | 100.8 |
| 84 | 28 | 37 | 73.9 | 8.7 | 49.3 | 40 | 159 | 236 | 28.4 | 37.1 | 60 | 100.7 |
| 74 | 28 | 35 | 73.7 | 8.2 | 48.2 | 40 | 159 | 236 | 28.4 | 36.7 | 59 | 100.6 |
| 1 | 33 | 39 | 80.2 | 12.3 | 62.8 | 28 | 171 | 253 | 30.5 | 42.8 | 68 | 103.3 |
| 64 | 34 | 35 | 80.6 | 11.6 | 62.3 | 26 | 173 | 257 | 30.9 | 42.6 | 68 | 103.3 |
| 63 | 33 | 35 | 79.5 | 11.0 | 59.8 | 28 | 171 | 253 | 30.5 | 41.5 | 66 | 102.8 |
| 6 | 31 | 35 | 77.1 | 9.8 | 54.9 | 33 | 166 | 246 | 29.7 | 39.5 | 63 | 101.9 |
| 262 | 30 | 45 | 77.5 | 11.9 | 59.1 | 35 | 164 | 243 | 29.3 | 41.2 | 66 | 102.6 |
| 16 | 27 | 43 | 73.5 | 9.6 | 50.4 | 42 | 157 | 233 | 28.0 | 37.6 | 60 | 101.0 |
| 58 | 30 | 55 | 79.1 | 14.7 | 65.5 | 35 | 164 | 243 | 29.3 | 43.9 | 70 | 103.8 |
| 94 | 28 | 55 | 76.4 | 13.0 | 59.7 | 40 | 159 | 236 | 28.4 | 41.5 | 66 | 102.8 |
| 60 | 27 | 54 | 74.9 | 12.0 | 56.3 | 42 | 157 | 233 | 28.0 | 40.1 | 64 | 102.1 |
| 4 | 26 | 54 | 73.5 | 11.3 | 53.7 | 44 | 155 | 229 | 27.6 | 38.9 | 62 | 101.6 |

The calculation is done for the worst-case scenario for adiabatic cooling when the combustion air does not consider the additional humidity from water spraying. This means that the water spray nozzles are not installed in the way of the combustion air, or the combustion air is taken from the outside room. However, in operation, the total separation of the two systems is possible only if combustion air is provided with air ducts from outside. In the case of ships that take combustion air from the engine room, total separation will not be possible. Therefore, the efficiency of the direct adiabatic cooling will increase because part of the combustion air flow will take part in humidity. Increasing the relative humidity of the combustion air will affect the combustion system, with the advantages and disadvantages highlighted in Section 2.4 from above.

**Table 9.** Direct Adiabatic cooling power calculated for cooling air flow of 12,350 m$^3$/h (2 fans of 13.000 m$^3$/h, total air flow of 26.000 m$^3$/h), inside air temperature 45 °C.

| | Outside Air | | | | | Cooling Power | | | Inside Air after Cooling | | |
|---|---|---|---|---|---|---|---|---|---|---|---|
| Hours | Temperature | RH | THI—Temp Humidity Index | Absolute MOISTURE | Enthalpy | Sensible Cooling Power | Adiabatic Cooling (Latent Heat) | Water Flow | Absolute Moisture After Direct Adiabatic Cooling | RH | THI—Temp Humidity Index |
| 01 | 02 | 03 | 04 | 05 | 06 | 07 | 08 | 09 | 11 | 12 | 13 |
| | [°C] | (%) | | (g/kg) | (kJ/kg) | [kw] | [kw] | [L/h] | (g/kg) | (%) | |
| 1 | 28 | 38 | 74.1 | 8.9 | 49.9 | 67 | 132 | 196 | 23 | 38 | 94.2 |
| 84 | 28 | 37 | 73.9 | 8.7 | 49.3 | 67 | 132 | 196 | 23 | 38 | 94.2 |
| 74 | 28 | 35 | 73.7 | 8.2 | 48.2 | 67 | 132 | 196 | 22 | 36 | 93.7 |
| 1 | 33 | 39 | 80.2 | 12.3 | 62.8 | 47 | 152 | 225 | 28 | 46 | 96.5 |
| 64 | 34 | 35 | 80.6 | 11.6 | 62.3 | 43 | 156 | 231 | 28 | 46 | 96.5 |
| 63 | 33 | 35 | 79.5 | 11.0 | 59.8 | 47 | 152 | 225 | 27 | 44 | 96.0 |
| 6 | 31 | 35 | 77.1 | 9.8 | 54.9 | 55 | 144 | 214 | 25 | 41 | 95.1 |
| 262 | 30 | 45 | 77.5 | 11.9 | 59.1 | 59 | 140 | 208 | 27 | 44 | 96.0 |
| 16 | 27 | 43 | 73.5 | 9.6 | 50.4 | 70 | 129 | 190 | 23 | 38 | 94.2 |
| 58 | 30 | 55 | 79.1 | 14.7 | 65.5 | 59 | 140 | 208 | 30 | 35 | 93.2 |
| 94 | 28 | 55 | 76.4 | 13.0 | 59.7 | 67 | 132 | 196 | 27 | 44 | 96.0 |
| 60 | 27 | 54 | 74.9 | 12.0 | 56.3 | 70 | 129 | 190 | 26 | 42 | 95.6 |
| 4 | 26 | 54 | 73.5 | 11.3 | 53.7 | 74 | 125 | 185 | 25 | 41 | 95.1 |

The final absolute moisture after water spraying is calculated considering 100% water vaporization. Based on the final moisture in the air, the final relative humidity is determined from the Mollier chart, and then the THI index is calculated.

In Table 8, the calculation is done for the minimum air flow requested by the ISO standard (150% from combustion air flow). The waterflow for the adiabatic cooling is calculated to ensure a maximum temperature of 45 °C in the engine room. Combustion air flow is not included; therefore, only the cooling air flow of 7350 m$^3$/h is used in the adiabatic cooling calculations.

The maximum relative humidity of RH60%, as recommended by the International Association of Classification (IACS) [11,12], and the THI index are checked. A temperature of 45 °C and RH60% are accepted for good functionality of the equipment, but without people inside, taking into consideration the temperature-humidity index, which is over 100 in these environmental conditions. According to the calculation results, it can be concluded that in alternative 1, the ventilation system can assure the necessary combustion air and cooling of the room by running the ventilation system at 31%, but the relative humidity will be over 60%. These conditions are not recommended by the IACS. The temperature-humidity index is over 100, which means that people cannot live inside because the human body overheats and leads to death.

In the second alternative (Table 9), the air flow is increased to 38% by starting the two fans of 13.000 m$^3$/h. As can be seen in Table 9, the cooling power and inside environ-

mental conditions are improved, and the system is able to assure the necessary cooling and combustion air, also improving the value of the comfort index inside the room, as indicated in Table 9. These environmental conditions are accepted for the good functionality of the equipment and for human life, but the THI index is over 90, which means the discomfort state.

In these conditions, the adiabatic cooling system should be supplied with about 200 L/h of clean water to stop 4 fans with total electrical power of about 8.8 kW).

From an energy point of view, the ventilation system with adiabatic cooling is more efficient because in a high-pressure water system of 100 bar, the energy consumption for the water pump with an efficiency of about 50% will be about 0.6 kWh and maximum 2.5 kWh for water cleaning if the reverse osmosis system is used versus about 8.8 kWh needed for running the fans.

Therefore, by using direct adiabatic cooling, the power consumption of the ventilation system can be reduced by up to about 5 kW for the reference vessel. Then, for a total time of 943 h in summertime, the decrease in electrical energy can be about 4700 kWh. Considering the average value of fuel consumption of 230 g/kW [8,9] for diesel generators, the total fuel retrenchment will be about 1250 L.

The adiabatic cooling system can be provided with an automatic start stop based on the temperature and relative humidity sensors installed inside the engine room.

## 4. Conclusions

The direct adiabatic cooling system can be used in addition to the existing ventilation systems for cooling the engine room in case the existing ventilation systems cannot ensure cooling requirements when outside air is hot. It is most suitable for unattended machinery spaces. According to the calculation carried out, the capacity of the ventilation system can be reduced to 38% by using direct adiabatic cooling, and power consumption can be reduced by over 50% in the reference vessel. This system is based on increasing the relative humidity inside the engine room, which will have the following negative impact:

- A small decrease in the efficiency of the combustion is expected, but this will not affect the power while the engine is designed to work at 45 °C and 60%RH.
- High humidity can generate condensation in the air cooler due to the high pressure after the turbocharger and dropping down the temperature inside the cooler, but only if the combustion air is part of direct adiabatic cooling.
- The comfort of the crew inside the engine room will be reduced by increasing the temperature humidity comfort index.

Combustion efficiency is not affected, and the condensation issue can be avoided if the combustion air system is provided with an outside air intake, according to the Caterpillar [35] recommendations.

Regarding the positive impact, the following can be mentioned:

- The air flow can be substantially reduced, which means reducing the power consumption and pressure drop across the ventilation ducts and louvers, which will also reduce the noise from the ventilation system.
- Reducing temperature and increasing humidity will have a positive effect by substantially reducing pollution (NOx).
- Reducing the air flow will reduce the air velocity inside the engine room, which should reduce the heat dissipation to the air from the engines. More energy will be removed by the internal water cooling system of the engines.
- Regarding the airborne spreading of viruses, increasing the relative humidity will reduce the degree of infectivity.

It is recommended that the water used in adiabatic cooling be cleaned so that the minerals are reduced below 90 ppm and the bacteria are removed by chemical or UV cleaning systems. The fresh water at the local suppliers has a quantity of minerals between

195 and 658 ppm. Therefore, a cleaning or softening system should be used to reduce the hardness of the water used for direct adiabatic cooling to 90 ppm.

Considering that during adiabatic cooling, the temperature humidity index (THI) is increasing over 80, which is considered the discomfort state, direct adiabatic cooling is not recommended in the Danube area for vessels with attended engine rooms. The solution can be used in periodically unattended machinery spaces and in cases of extreme environmental conditions to avoid reducing the capability of the ship.

## 5. Index of Notations and Abbreviations

| Acronym | Meaning |
| --- | --- |
| DCF | Dry exhaust and turbo manifold correction factor |
| Ter | Ambient Engine Room temperature |
| Qa/Qb | The air flow needed for cooling and for combustion (alternative a/b) |
| qh | The air flow needed for cooling |
| qc | The air flow needed for combustion |
| qha | The air flow of ventilation system without combustion air |
| P | The engine power |
| ρ | The air density |
| c | The specific heat capacity of the air |
| ΔT | Increasing of the air temperature in engine room (according to ISO) |
| ΔTcs | Difference of temperature between outside air and inside air |
| Δh | The difference of air mixture (air &water vapor) enthalpy |
| ΣΦ | The sum of heat radiation of equipment inside engine room |
| Φm | Cooling power for minimum requested air flow |
| Φc | Cooling power for combustion air flow |
| Φcs | Sensible cooling power of cooling air |
| Φl | The cooling capacity of water spraying (latent heat exchange) |
| Φt | The total cooling capacity (latent & sensible heat exchange) |
| qw | The water flow for adiabatic cooling |
| ρw | The water density |
| Hva | The heat of vaporization of water |
| THI | The temperature humidity index |
| T | The air temperature |
| U | The Relative humidity |
| ppm | Parts per million |
| UV | Ultraviolet |
| HP | Horsepower |
| IACS | International Association of Classification Societies |
| ISO | International Organization for Standardization |
| CFD | Computational Fluid Dynamics |
| CIMAC | International Council on Combustion Engines |

**Author Contributions:** Conceptualization, investigation, visualization, methodology and writing—original draft preparation, V.M.; supervision, formal analysis, writing—review and editing, L.R. All authors have read and agreed to the published version of the manuscript.

**Funding:** This work was carried out in the framework of the research project CLIMEWAR (CLimate change IMpact Evaluation on future WAve conditions at Regional scale for the Black and Mediterranean seas marine system), supported by a grant from the Ministry of Research, Innovation and Digitization, CNCS—UEFISCDI, project number PN-III-P4-PCE-2021-0015, within PNCDI III.

**Institutional Review Board Statement:** Not applicable.

**Informed Consent Statement:** Not applicable.

**Data Availability Statement:** Publicly available datasets were analyzed in this study. This data can be found here: see references [26–29,33].

**Acknowledgments:** This work was also presented at the 10th edition of the Scientific Conference organized by the Doctoral Schools of "Dunărea de Jos" University of Galati (SCDS-UDJG) http://www.cssd-udjg.ugal.ro/ (accessed on 10 July 2022), which was held on 9–10 June 2022, in Galati, Romania.

**Conflicts of Interest:** The authors declare no conflict of interest.

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
