# Peer review of "Improving the Ventilation of Machinery Spaces with Direct Adiabatic Cooling System"

_inventions, doi:10.3390/inventions7030078_

Round 1

Reviewer 1 Report

This is an interesting piece of work and I enjoyed reading it.

Although evaporative/adiabatic cooling is commonly in use, its application in engine rooms on vessels is somewhat innovative. Therefore, in my opinion this work is within scope of the inventions journal.

The paper layout is adequate however the distribution needs attention. Section 2.7 can be condensed considerably by omitting some (repetitive) details. The x-axes of the graphs in this section need to be organised better, by including date and consistent daily timestamps. Additionally, it is recommended to remove graph titles and incorporate them in the captions. Figure 6 is not referenced in the text, thus unnecessary. The data in Figures 6, 9, 12 & 15 is misleading since they give the impression that the temperature never goes below the set lower limit of 25/30oC.   

Please make sure that SI units are used throughout the whole work.

It seems that in page 3 Line 42 qc is wrongly defined.

With reference to the air flow requirements defined on page 3 lines 86-87 and Table 3, please explain why in page 4 line 118 the minimum air flow of 20475m3/h is considered rather than the value of 62710m3/h.

Please advise how you arrived at the fan flow rate values (page 4 lines 110-112) by stating whether these were measured values or whether they refer to free flow values.

THI equation on page 4 line 145 is more legible if the term (tx1.8 + 32) is written as (1.8T+32).

Table 6 has the wrong units for energy (W/kg).

The equation on page 15 lines 345-352 has inconsistent units.

Table 7 is quite confusing. Please rearrange the data presentation and explain it further in text.

The results and discussion section needs to be more detailed and improved.

I suggest keeping the layout of Tables 9 and 10 the same. These Tables need to be explained in more detail in the text. 

Author Response

Response to Reviewer 1 Comments

Manuscript ID: inventions-1871457

GENERAL COMMENTS

A revision has been carried out following carefully all the indications, suggestions, and observations formulated by the reviewers.

The main changes operated are outlined next:

1) Detailed explanations and additional information have been added at various points to make the results of the work more comprehensible.

2) Some tables were updated

3) Section 2.7 has been synthesized to reduce its size, according to the observation received from reviewers

At this point, it must be also highlighted that the authors tried to follow all the suggestions and observations formulated by the reviewers and to operate (as much as it was possible) all the corrections indicated by them. Furthermore, to follow the corrections operated in the paper, a version of the manuscript having all the changes operated tracked (with the option track changes) has been also uploaded together with the last form of the manuscript (without the changes tracked).

The specific corrections operated according to the suggestions of the reviewers are given next together with detailed explanations.

Point 1: This is an interesting piece of work and I enjoyed reading it.

Although evaporative/adiabatic cooling is commonly in use, its application in engine rooms on vessels is somewhat innovative. Therefore, in my opinion this work is within scope of the inventions journal.

Response 1: Thank you for your considerations and appreciation of our study

Point 2: The paper layout is adequate however the distribution needs attention. Section 2.7 can be condensed considerably by omitting some (repetitive) details. The x-axes of the graphs in this section need to be organised better, by including date and consistent daily timestamps. Additionally, it is recommended to remove graph titles and incorporate them in the captions. Figure 6 is not referenced in the text, thus unnecessary. The data in Figures 6, 9, 12 & 15 is misleading since they give the impression that the temperature never goes below the set lower limit of 25/30oC.   

Response 2: The repetitive graphs for different months in section 2.7 were replaces with a short descriptive conclusion. Furthermore, the graph title was incorporated in the figure according to the suggestion of the reviewer.

Point 3: Please make sure that SI units are used throughout the whole work.

Response 3: The paper was checked and additional information was added, where found necessary.

Point 4: It seems that in page 3 Line 42 qc is wrongly defined.

Response 4: The above text was corrected (the new text is “the air flow needed for combustion”)

Point 5: With reference to the air flow requirements defined on page 3 lines 86-87 and Table 3, please explain why in page 4 line 118 the minimum air flow of 20475m3/h is considered rather than the value of 62710m3/h.

Response 5: In case when the cooling is done using the ventilation system only, the total air flow needed for cooling and combustion is 62710 m3/h. If the cooling of the room is done using other systems (e.g., direct adiabatic cooling, fan coils etc) the ventilation system must ensure, as a minimum requirement, 150% from the air flow requested for combustion, which is 20475m3/h.

Point 6: Please advise how you arrived at the fan flow rate values (page 4 lines 110-112) by stating whether these were measured values or whether they refer to free flow values.

Response 6: The number and capacity of the fans indicated in section  2.2 are according to the technical data checked on board of the vessel. The air flow was not measured, but the technical data found on board is maching with the calculation of the engine room ventilation included in this study.

Point 7: THI equation on page 4 line 145 is more legible if the term (tx1.8 + 32) is written as (1.8T+32).

Response 7: The formula was updated according to sudgestion of the reviewer.

Point 8: Table 6 has the wrong units for energy (W/kg).

Response 8: The description of table 4 was updated. The table presents the heat of water vaporization per water mass used.

Point 9: The equation on page 15 lines 345-352 has inconsistent units.

Response 9: Thank you for this observation, the units are updated to comply with the formula.

Point 10: Table 7 is quite confusing. Please rearrange the data presentation and explain it further in the text.

Response 10: The data were rearranged in the table to be clearer. In addition, an explanation note was added.

Table 7. Time summary of relative humidity for temperatures above 25°C during June, July, August and September, 2021.

Relative humidity

All

RH<40%

40%≥RH≤50%

50%>RH<60%

RH≥60%

Temperature

≥25°C

<30°C

≥30°C

≥25°C

≥25°C

<30°C

≥30°C

June [hours]

(Average temp. & RH)

164

-

1

(28°C,38%RH)

1

(33°C,39%RH)

56

(30°C,45%RH)

58

(30°C,55%RH)

45

3

July [hours]

(Average temp. & RH)

337

-

22

(28°C,37%RH)

64

(34°C,35%RH)

113

(30°C,45%RH)

94

(28°C,55%RH)

44

0

August [hours]

(Average temp. & RH)

342

-

62

(28°C,37%RH)

63

(33°C,35%RH)

93

(30°C,45%RH)

60

(27°C,54%RH)

62

2

September [hours]

(Average temp. & RH)

100

-

74

(28°C,35%RH)

6

(31°C,35%RH)

16

(27°C,43%RH)

4

(26°C,54%RH)

0

0

TOTAL [hours]

943

159

134

278

216

151

5

16.9%

14.2%

29.5%

22.9%

16.0%

0.5%

293

278

216

156

31.1%

29.5%

22.9%

16.5%

Note: The data indicate in the brackets “()” represents the average temperature and relative humidity according to recorded data (e.g. in June the RH<40%, and temp < 30°C was recorded for 22 hours, and the average temperature and relative humidity in this time was 28°C,37%RH). The average temperature and relative humidity will be used for future calculation and analyzes.

Point 11: The results and discussion section needs to be more detailed and improved.

I suggest keeping the layout of Tables 9 and 10 the same. These Tables need to be explained in more detail in the text. 

Response 11: Table 7 was updated with some information to be easier to understand.

In the chapter “3. Results and Discussions” we added the following paragraph:

“In the first alternative the cooling capacity of the ventilation system with direct adiabatic cooling is calculated for the minimum requested air flow, by starting one fan of 13.000m3/h and one fan of 8000m3/h (total 21.000m3/h).

In the second alternative, the air flow is increased so that the requested cooling capacity to be covered in all outside environmental conditions, without exceeding the relative humidity of 60%. In this calculation two fans of 13.000m3/h (total 26.000m3/h), are used.

In Tables 9 and 10 the columns 01, 02, and 03 are according to Table 7, based on the data recorded. The temperature and humidity index indicated in columns 4 and 13 are calculated according to formula (8) for outside air and for inside air after water spraying.

The absolute moisture (column 05), enthalpy (column 06) and final relative humidity (column 12) are determined from Mollier Chart [34]. The sensible cooling power of the cooling air (column 07) is calculated according to formula (7.1). Adiabatic cooling latent heat (column 08) is calculated as the difference between total cooling power needed (199kW) and sensible cooling power calculated in column 07. The water flow for adiabatic cooling (column 09) is calculated with formula (9.1) using the latent cooling power from column 08. The absolute moisture added by the adiabatic cooling (column 10) is calculated according to water flow in (column 09) and air flow for cooling. The absolute moisture after the direct adiabatic cooling is calculated as a sum of column 5 and column 10 and based on this the final relative humidity is determined from Mollier chart.

It should be pointed out that the calculation is done for the worst-case scenario for adiabatic cooling, when the combustion air is not taking the additional humidity from water spraying. This means the water spray nozzles are not installed in the way of the combustion air or the combustion air is taken from outside the room. However, in operation the total separation of the two systems is possible only if the combustion air is provided with air ducts from outside. In case of ships which take the combustion air form the engine room the total separation will not be possible. Therefore the efficiency of the direct adiabatic cooling will increase because a part of combustion air flow will take a part of humidity. Increasing the relative humidity of the combustion air will affect the combustion system, with advantages and disadvantages highlighted in section 2.4 above.

The final absolute moisture after water spraying is calculated considering 100% water vaporization. Based on the final moisture in the air, the final relative humidity is determined from Mollier chart and then THI index is calculated.”

In the Section “3. Results and Discussions” the following paragraph was updated:

“In Table 9 below the calculation is done for the minimum air flow requested by ISO standard (150% from combustion air flow) for the highest temperature of 45°C. The waterflow for adiabatic cooling is calculated to ensure the temperature of maximum 45°C in the engine room. The combustion air flow is not included in the calculations, therefore only the cooling air flow of 7350m3/h is used in adiabatic cooling calculations.

The maximum relative humidity of RH60%, as recommended by the International Association of Classification (IACS) [11] [12], and the THI index are checked. The temperature of 45°C and RH60% are accepted for a good functionality of the equipment, but without people inside, taking into consideration the temperature-humidity index which is over 100 in these environmental conditions. According to the calculation results, it can be concluded that in alternative 1 the ventilation system can assure the necessary combustion air and cooling of the room by running the ventilation system at 31% but the relative humidity will be over 60% some times. These conditions are not recommended by IACS. The temperature-humidity index is over 100, which means that the people cannot work inside because the human body overheats which leads to death.

In Section  “3. Results and Discussions” Tables 9 and 10 are updated to have the same information and format. Furthermore, some data are corrected in Table 10.

Table 9. THI index, relative humidity and Direct Adiabatic cooling power calculated for cooling air flow of 7350m3/h and total air flow of 21.000 m³/h (one fan of 8000 and one of 13.000m³/h), inside air temperature 45°C, RH according to calculation

Outside air

Cooling power

Inside air

Hours

Temperature

RH

THI- Temp Humidity index

Absolute Moisture

Enthalpy

Sensible cooling power

Adiabatic cooling
(latent heat)

Water flow

Moisture added

 Absolute Moisture after direct adiabatic cooling

RH

THI- Temp Humidity index

01

02

03

04

05

06

07

08

09

10

11

12

13

[°C]

(%)

(g/kg)

(kJ/kg)

[kw]

[kw]

[l/h]

(g/kg)

(g/kg)

(%)

(%)

1

28

38

74.1

8.9

49.9

40

159

236

28.4

37.4

60

100.8

84

28

37

73.9

8.7

49.3

40

159

236

28.4

37.1

60

100.7

74

28

35

73.7

8.2

48.2

40

159

236

28.4

36.7

59

100.6

1

33

39

80.2

12.3

62.8

28

171

253

30.5

42.8

68

103.3

64

34

35

80.6

11.6

62.3

26

173

257

30.9

42.6

68

103.3

63

33

35

79.5

11.0

59.8

28

171

253

30.5

41.5

66

102.8

6

31

35

77.1

9.8

54.9

33

166

246

29.7

39.5

63

101.9

262

30

45

77.5

11.9

59.1

35

164

243

29.3

41.2

66

102.6

16

27

43

73.5

9.6

50.4

42

157

233

28.0

37.6

60

101.0

58

30

55

79.1

14.7

65.5

35

164

243

29.3

43.9

70

103.8

94

28

55

76.4

13.0

59.7

40

159

236

28.4

41.5

66

102.8

60

27

54

74.9

12.0

56.3

42

157

233

28.0

40.1

64

102.1

4

26

54

73.5

11.3

53.7

44

155

229

27.6

38.9

62

101.6

In Section “3. Results and Discussions” we added also the following paragraph:

“Therefore, by using the direct adiabatic cooling” the power consumption of the ventilation system can be reduced up to about 5kWh for the reference vessel. Then, for a total time of 943 hours in summer time, the reducing of the electrical power can be about 4700kW. Considering the average value of fuel consumption of 230g/kW [8-9] for diesel generators, the total fuel retrenchment will be about 1250 liters.”

Thank you again for your feedback. We hope that the changes made and our answer meet the requirements expressed in the observations formulated.

Reviewer 2 Report

·         The title should be modified and give a complete understanding of the text to the audience.

·         The Abstracts must contain at least 150 words up to 250 words, consist of

2-3 sentences as brief intro about the paper, 2-3 sentences to describe how

the problem is solved, and 2-4 sentences showing the results of experiments/simulation, ended with 1-2 sentences as short main conclusions of the work.

·         Add abbreviations, Nomenclatures and Subscript.

·         The Introduction section must explain the background of the problem and the urgency of the study, which can be proved by providing some previous researches and works, and also how to solve the problem in brief.

·         In the introduction, long texts do not have reference, references must be mentioned for each sentence.

·         The main objective of the work must be written in a clearer and more concise way at the end of the introduction section.

·         Add references to all formulas and tables 1-3.

·         For conclusions, more detailed results should be presented.

·          Better description and explanation of results.

·         Update reference 2018 to 2022.

Author Response

Response to Reviewer 1 Comments

Manuscript ID: inventions-1871457

GENERAL COMMENTS

A revision has been carried out following carefully all the indications, suggestions, and observations formulated by the reviewers.

The main changes operated are outlined next:

1) Detailed explanations and additional information have been added at various points to make the results of the work more comprehensible.

2) Some tables were updated

3) Section 2.7 has been synthesized to reduce its size, according to the observation received from reviewers

At this point, it must be also highlighted that the authors tried to follow all the suggestions and observations formulated by the reviewers and to operate (as much as it was possible) all the corrections indicated by them. Furthermore, to follow the corrections operated in the paper, a version of the manuscript having all the changes operated tracked (with the option track changes) has been also uploaded together with the last form of the manuscript (without the changes tracked).

The specific corrections operated according to the suggestions of the reviewers are given next together with detailed explanations.

Point 1: The title should be modified and give a complete understanding of the text to the audience.

Response 1: The title proposed is:

Improving the Ventilation of Machinery Spaces with Direct Adiabatic Cooling system"

Point 2:    The Abstracts must contain at least 150 words up to 250 words, consist of

2-3 sentences as brief intro about the paper, 2-3 sentences to describe how

the problem is solved, and 2-4 sentences showing the results of experiments/simulation, ended with 1-2 sentences as short main conclusions of the work.

Response 2: The Abstract was updated:

“Abstract: The machinery spaces are provided with ventilation systems which should assure the necessary airflow for combustion and for cooling. In some vessels, due to the space constrains the requested air flow for cooling cannot be achieved in extreme environmental conditions and the engines load should be reduced. On the other hand, the outside air temperature can increase over 35°C in some places and the efficiency of the ventilation is reduced. In these cases, other solutions for cooling of engine room should be analyzed. In this paper the environmental conditions in the Romanian Danube area are analyzed to understand if the direct adiabatic cooling can be used to improve the ventilation system and what are the challenges after increasing the relative humidity inside the machinery spaces. Based on the data recorded, it is noted that the outside relative humidity is highly dropping down when the temperature is increasing which will ensure the good condition for using of the adiabatic cooling. This study demonstrates that by using a direct adiabatic cooling, the air flow of the ventilation system can be reduced with more that 50%, which will reduce the pressure drop across the ventilation system, the noise and energy consumption. After the adiabatic cooling, the temperature and relative humidity inside the engine room will be proper for functionality of the equipment but the temperature-humidity index will be high which means high discomfort for the crew. Therefore, it is concluded that this cooling solution should be used in unattended machinery spaces only.

Point 3:  Add abbreviations, Nomenclatures and Subscript.

Response 3: Thank you for your suggestion and index of nottations and abbreviations has been included.

Point 4:  The Introduction section must explain the background of the problem and the urgency of the study, which can be proved by providing some previous researches and works, and also how to solve the problem in brief.

Response 4: In the chapter “1. Introduction” we updated the following paragraph:

Taking into consideration that in the last time the size of the engine room was reduced and the total power was increased, some compromises are done during the detailed design and construction process of the vessel due to the space constrains [2]. Consequently, the pressure drop across the ventilation systems can increase above the fan capacity and finally the air flow will be lower than the cooling requirements. Sometimes the issue is discovered in the late stage of the project, during sea trial or in operation. In this stage of the project in general it is almost impossible to reduce the pressure drop. Therefore, other solutions are necessary to improve the ventilation system, like Hendri et al. [3] which have performed CFD analyses in relationship with changing the air distribution and adding new fans. Unfortunately, this kind of improvement is expensive, time consuming and sometimes is asking for big changes on board which is not always possible. In other situation the Owner is looking for an easy and cheaper alternative, therefore other solutions should be analyzed.  One solution which can solve this issue is the direct adiabatic cooling which is easy to be installed, with low energy consumption but it could have a negative impact to equipment and to people working inside, if the relative humidity is increasing too much. The efficiency of the system must be analyzed for each area based on the expected temperatures and relative humidity of the outside air.”

Point 5:  In the introduction, long texts do not have reference, references must be mentioned for each sentence.

Response 5: Following the above suggestion of the reviewer the references have been indicated.

Point 6:  The main objective of the work must be written in a clearer and more concise way at the end of the introduction section.

Response 6: In Sectiom “1. Introduction” we added the following paragraph:

“The main objectives of the study are to check if the direct adiabatic cooling is feasible to be used in engine room ventilation and what are the challenges, constrains and benefits for adding this cooling type to engine room ventilation system.”

Point 7: Add references to all formulas and tables 1-3.

Response 7: Referenece were added for all fromulas and Tables 1-3

Point 8: For conclusions, more detailed results should be presented.

Response 8: Section “3. Results and Discussions” was updated with more informationaccording to response 9.

In Section “4. Conclusions” the first paragraph was updated.

Point 9: Better description and explanation of results.

Response 9: Table 7 was updated with some information to be easier to understand.

In Section “3. Results and Discussions” we added the following paragraph:

“In the first alternative the cooling capacity of the ventilation system with direct adiabatic cooling is calculated for the minimum requested air flow, by starting one fan of 13.000m3/h and one fan of 8000m3/h (total 21.000m3/h).

In the second alternative, the air flow is increased so that the requested cooling capacity to be covered in all outside environmental conditions, without exceeding the relative humidity of 60%. In this calculation two fans of 13.000m3/h (total 26.000m3/h), are used.

In Tables 9 and 10 the columns 01, 02, and 03 are according to table 7, based on data recorded. The temperature and humidity index indicated in column 4 and 13 are calculated according to formula (8) for outside air and for inside air after water spraying.

The absolute moisture (column 05), enthalpy (column 06) and final relative humidity (column 12) are determined from Mollier Chart [34]. The sensible cooling power of the cooling air (column 07) is calculated according to formula (7.1). Adiabatic cooling latent heat (column 08) is calculated as the difference between total cooling power needed (199kW) and sensible cooling power calculated in column 07. The water flow for adiabatic cooling (column 09) is calculated with formula (9.1) using the latent cooling power from column 08. Absolute moisture added by adiabatic cooling (column 10) is calculated according to water flow in (column 09) and air flow for cooling. The absolute moisture after direct adiabatic cooling is calculated as a sum of column 5 and column 10 and based on this the final relative humidity is determined from Mollier chart.

It should be pointed out that the calculation is done for the worst-case scenario for adiabatic cooling, when the combustion air is not taking the additional humidity from water spraying. That means the water spray nozzles are not installed in the way of the combustion air or the combustion air is taken from outside the room. However, in operation the total separation of the two systems is possible only if the combustion air is provided with air ducts from outside. In case of ships which take the combustion air form engine room the total separation will not be possible therefore the efficiency of direct adiabatic cooling will increase because a part of combustion air flow will take a part of humidity. Increasing the relative humidity of the combustion air will affect the combustion system, with advantages and disadvantages highlighted in chapter 2.4 above.

The final absolute moisture after water spraying is calculated considering 100% water vaporization. Based on final moisture in the air, the final relative humidity is determined from Mollier chart and then THI index is calculated.”

In Section “3. Results and Discussions” the following paragraph was updated:

“In Table 9 below the calculation is done for minimum air flow requested by ISO standard (150% from combustion air flow) for the highest temperature of 45°C. The waterflow for adiabatic cooling is calculated to ensure the temperature of max 45°C in engine room. The combustion air flow if not included in the calculations therefore only the cooling air flow of 7350m3/h is used in adiabatic cooling calculations.

The maximum relative humidity of RH60%, as recommended by the International Association of Classification (IACS) [11] [12], and the THI index are checked. The temperature of 45°C and RH60% are accepted for a good functionality of the equipment, but without people inside, taking into consideration the temperature-humidity index which is over 100 in these environmental conditions. According to the calculation results, it can be concluded that in alternative 1 the ventilation system can assure the necessary combustion air and cooling of the room by running the ventilation system at 31% but the relative humidity will be over 60% some times. These conditions are not recommended by IACS. The temperature-humidity index is over 100 which means that the people cannot work inside because the human body overheats which leads to death.

In Section “3. Results and Discussions” the table 9 and 10 are updated to have the same information and format, according to sudgestion form another reviwer. Some datas are corrected in table 10.

In Section “3. Results and Discussions” we added the following paragraph:

“Therefore, by using direct adiabatic cooling” the power consumption of the ventilation system can be reduced up to about 5kWh for reference vessel. Then, for total time of 943 hours in summer time, the reducing of the electrical power can be about 4700kW. Considering the average of fuel consumption of 230g/kW [8-9] for diesel generators the total fuel retrenchment will be about 1250 liters.”

Point 10: Update reference 2018 to 2022.

Response 10: We are assuming that the comment is reffering to reference 3 and 15.

We would like to mention that the reffrence 3 is indicating the possibility to improve the ventilation system by CFD analises and this paper is a good and proven project. The main purpose of this reference is to indicate that by using CFD analisysis the ventilation can be improved. However, some times this kind of improvement is expensive and the Owner is looking for an easier and simpler solution which could be direct adiabatic cooling.

Regarding to reference 15 this is used for calculation of the temperature-humidity index only.

[3]          N. S. Hendri, A. Adi, O. S. Suharyo, and A. D. Susanto, “The Air Flow Analysis in Engine Rooms at Frigate Class Ship with CFD Approach (Computational Fluids Dynamics),” Int. J. Recent Eng. Sci., vol. 5, no. 4, pp. 11–18, 2018, doi: 10.14445/23497157/ijres-v5i4p103.

[15]        Termotecnica, “Temperature Humidity Index: what you need to know about it,” 2018. https://www.pericoli.com/EN/news/120/Temperature-Humidity-Index-what-you-need-to-know-about-it.html

Round 2

Reviewer 1 Report

The paper is much more organised and readable. Yet the following recommendations should be considered.

Please explain what the third column in Table 4 refers to (it seems t have wrong units). 

In Table 7 the temperature ranges are inconsitent for the various RH. The reviewer suggest to use consistent temp ranges (>=30 & <30).

In Section 3 results and Discussion the author is confusing power with energy in the added paragraph at the end. It should read

"Therefore, by using the direct adiabatic cooling the power consumption of the ventilation system can be reduced by up to about 5kW for the reference vessel. Then, for a total time of 943 hours in summer time, the reducing of the electrical energy can be about 4700kWh. Considering the average value of fuel consumption of 230g/kW [8-9] for diesel generators, the total fuel retrenchment will be about 1250 litres."

The English language needs attention.

Regards.

Author Response

Point 1: The paper is much more organised and readable. Yet the following recommendations should be considered.

Response 1: Thank you very much for appreciating our work. Thank you also for your very constructive and useful comments and suggestions!

Point 2: Please explain what the third column in Table 4 refers to (it seems t have wrong units). 

Response 2: Thank you for this observation. The wrong unit for energy was corrected (Wh/kg instead of W/kg).

Point 3: In Table 7 the temperature ranges are inconsitent for the various RH. The reviewer suggest to use consistent temp ranges (>=30 & <30).

Response 2: Table 7 was updated according to the reviewer's suggestion (the temperature range of „>=30 & <30” was used)

Point 4: In Section 3 results and Discussion the author is confusing power with energy in the added paragraph at the end. It should read

"Therefore, by using the direct adiabatic cooling the power consumption of the ventilation system can be reduced by up to about 5kW for the reference vessel. Then, for a total time of 943 hours in summer time, the reducing of the electrical energy can be about 4700kWh. Considering the average value of fuel consumption of 230g/kW [8-9] for diesel generators, the total fuel retrenchment will be about 1250 liters."

Response 4: Thank you for your observation. The text was updated according to your suggestions (the correct unit for power and description for energy were corrected).

Reviewer 2 Report

Accept

Author Response

Thank you!